# Epistemological Flexibility in Person-Centered Care: The Cynefin Framework for (Re)Integrating Indigenous Body Representations in Manual Therapy

**DOI:** 10.3390/healthcare12111149

**Published:** 2024-06-05

**Authors:** Rafael Zegarra-Parodi, Giandomenico D’Alessandro, Francesca Baroni, Jaris Swidrovich, Lewis Mehl-Madrona, Travis Gordon, Luigi Ciullo, Emiliano Castel, Christian Lunghi

**Affiliations:** 1BMS Formation, 75116 Paris, France; francesca@bms-formation.com (F.B.); christian@bms-formation.com (C.L.); 2Clinical-Based Human Research Department, Foundation Centre for Osteopathic Medicine (COME) Collaboration, 65121 Pescara, Italy; giandosteo89@hotmail.it; 3Research Department, A.T. Still Academy Italia (ATSAI), 70124 Bari, Italy; 4Leslie Dan Faculty of Pharmacy, University of Toronto, Toronto, ON M5S 3M2, Canada; jaris.swidrovich@utoronto.ca; 5Coyote Institute, Orono, ME 04473, USA; lewis.mehlmadrona@maine.edu; 6College of Osteopathic Medicine, Michigan State University, East Lansing, MI 48824, USA; gordont7@msu.edu; 7Istituto Europeo per la Medicina Osteopatica (IEMO), 16122 Genova, Italy; luigi.ciullo@univerosteo.it; 8Osteopathic Private Practice, 10249 Berlin, Germany; emiliano@escuelasomayalma.com

**Keywords:** epistemology, humanities, decision making, shared, problem-solving, native Americans, anthropology, medical, manual therapies, manipulation, osteopathic, primary prevention, secondary prevention, tertiary prevention

## Abstract

Background: Chiropractic, osteopathy, and physiotherapy (COP) professionals regulated outside the United States traditionally incorporate hands-on procedures aligned with their historical principles to guide patient care. However, some authors in COP research advocate a pan-professional, evidence-informed, patient-centered approach to musculoskeletal care, emphasizing hands-off management of patients through education and exercise therapy. The extent to which non-Western sociocultural beliefs about body representations in health and disease, including Indigenous beliefs, could influence the patient–practitioner dyad and affect the interpretation of pillars of evidence-informed practice, such as patient-centered care and patient expectations, remains unknown. Methods: our perspective paper combines the best available evidence with expert insights and unique viewpoints to address gaps in the scientific literature and inform an interdisciplinary readership. Results: A COP pan-professional approach tends to marginalize approaches, such as prevention-oriented clinical scenarios traditionally advocated by osteopathic practitioners for patients with non-Western sociocultural health assumptions. The Cynefin framework was introduced as a decision-making tool to aid clinicians in managing complex clinical scenarios and promoting evidence-informed, patient-centered, and culturally sensitive care. Conclusion: Epistemological flexibility is historically rooted in osteopathic care, due to his Indigenous roots. It is imperative to reintroduce conceptual and operative clinical frameworks that better address contemporary health needs, promote inclusion and equality in healthcare, and enhance the quality of manual therapy services beyond COP’s Western-centered perspective.

## 1. Introduction

Cultural norms, values, and traditions affect how bodies are depicted, viewed, and understood in the medical context. Sociocultural belief systems influence body representations in healthcare, perceptions of health, illness, disease, and subsequent treatment modalities [1]. These influences can manifest in diverse ways, including attitudes towards the perception of bodily sensations, the interpretation of symptoms in the context of a person’s life, and preference for certain treatment options. Understanding these sociocultural influences is crucial for providing culturally competent care, fostering patient trust, and addressing disparities in healthcare access and outcomes among different sociocultural groups [2]. Historically, Western biomedical models have influenced manual therapy practices that prioritize individualism, reductionism, and objectification of the body [3]. This Western lens tends to marginalize alternative healing traditions and Indigenous knowledge systems, perpetuating hierarchies of power and dominance that originate from colonial perspectives, which are also prevalent in healthcare [4]. The publication of the Flexner report in 1910 established the biomedical model as a new gold standard for medical education, which emphasized the study of science and technology [5] but resulted in the erasure of diverse cultural perspectives and the imposition of standardized approaches to healthcare [6]. 

In manual therapy, treatment modalities encompass various passive hands-on techniques such as massage, manipulation, and mobilization, and play a significant role in healthcare, particularly in managing musculoskeletal conditions. Non-medical professionals outside the United States, such as chiropractic, osteopathy, and physiotherapy (COP), have historically employed passive hands-on procedures aligned with their guiding principles. Non-medical COP research advocates evidence-informed, patient-centered musculoskeletal (MSK) care, emphasizing hands-off approaches such as education and exercise therapy [7,8,9]. Native American medicine has been practiced for at least 12,000 years in America. 

In Indigenous culture, the body is often considered sacred, representing the vehicle through which individuals experience a diverse range of human experiences and interact with the natural environment, community, and spiritual realms [10]. Symbolic imagery and metaphors are commonly employed to illustrate a body’s relationship with its cosmos, reflecting it as a vessel of ancestral wisdom transmitted across generations. These symbols shape healing practices centered on accessing and honoring wisdom to foster health and well-being [11]. Indigenous knowledge is transmitted orally through storytelling, myths, songs, and ceremonies. These oral traditions serve to preserve cultural beliefs and practices related to the body and their significance within the community [11]. Rituals, ceremonies, and practices are often employed to honor and protect the body’s sacredness, and this holistic view is understood within this specific sociocultural context.

Osteopathic care is a healthcare system that originated in the US in the late 19th century and has since been practiced and developed in various countries worldwide [12]. While osteopathic principles and practices used to guide clinical care are not tied to any culture or ethnicity [13], they have incorporated a legacy of key traditional Native American healing principles and reflect unique philosophical perspectives and approaches to healthcare [14]. Concepts from cross-cultural research and medical anthropology were initially introduced in Western healthcare by Professor Arthur Kleinman to provide an alternative framework for addressing health issues that are not adequately covered by traditional biomedical solutions [15]. Indeed, the Kleinman model introduced three concepts: illness, which refers to the subjective experience of being unwell, including perceptions and feelings; disease, which is a biomedical aspect, focusing on diagnosis and treatment; and sickness, which involves the social and cultural implications of illness on individuals’ roles and interactions within society [15]. Such models emphasize the importance of considering not only the biomedical aspects of symptoms but also the cultural, psychological, and social dimensions of the patient’s illness experience for a comprehensive understanding of health and healthcare practices, which are also found in manual therapy [16,17,18,19,20]. The last update of osteopathic tenets in 2002 emphasized a holistic understanding and treatment of patients [21]: a person is a dynamic unity of body, mind, and spirit that is in a continuous self-regulative process to adapt to the environment and social context because of the interconnectedness of physical structure and function [21]. These three important tenets are inherited from Indigenous healing traditions [14] and are recognized within the realm of complementary and alternative medicine (CAM) epistemology [22]. However, they raise ethical concerns owing to the lack of a clear definition and clinical framework to guide patient care [23]. 

As outlined by Tuscano et al. [5], a faction of the osteopathic profession has historically sought to minimize the non-biological aspects of osteopathy—its philosophical, existential/spiritual, emotional, and Native American roots—in an attempt to align it with the needs of a changing healthcare sector that established the primacy of the biological component of health.

Osteopathic care itself is not inherently a cultural treatment, in the same sense as traditional healing practices or therapies that are deeply rooted in specific cultural beliefs, traditions, and values [12]. However, a critical exploration of its Indigenous roots may (1) foster culturally sensitive care that goes beyond the Western-centered COP perspective [24,25] and (2) facilitate an ecological person-centeredness approach that is intrinsically embedded in Native American Indigenous healing traditions [25]. As stated by Finkel et al. [26], the historical and cultural contexts within which treatment modalities have evolved often reflect colonial legacies, perpetuating Western perspectives and power dynamics. The extent to which this legacy can also be applied to manual therapy remains unknown. Although the pan-professional COP approach suits Western healthcare epistemological frameworks, it may overlook non-Western clinical approaches. Epistemological flexibility involves acknowledging the possibility that practitioners’ perceptions of knowledge may not align with those of patients, thereby emphasizing the necessity of establishing a mutual horizon of understanding [27]. By cultivating epistemic humility between Western and non-Western patients’ sociocultural health assumptions, practitioners can employ reflective clinical reasoning to better meet patients’ values and expectations and provide evidence-informed, person-entered, and culturally sensitive care [24]. 

This perspective paper highlights, through a decolonial lens, a collaborative initiative aimed at exploring the influence of non-Western sociocultural beliefs on body representations in health, disease, and illness, and their potential impact on patient–practitioner interactions in line with evidence-informed practices. We argue that embracing epistemological flexibility, capable of integrating Indigenous perspectives on body representations, could enhance patient-centered care, inclusivity, and healthcare quality in manual therapy beyond the constraints of current Western frameworks. Within the manual therapy field, this paper focuses on the value of acknowledging the diversity of body representations in clinical practice through the application of current osteopathic principles and provides an example of a clinical scenario.

## 2. Materials and Methods

To address the aims reported above and considering that no prior studies have specifically examined decolonial perspectives on body representations and their potential implications in manual therapy in general, and osteopathy in particular, a perspective paper was considered the most appropriate method to explore this research question and initiate professional debate. We recognize that this method may be less robust compared to other established methods such as common types of reviews. However, given the absence of specific literature in this area, we have invited experts in the healthcare field with Indigenous backgrounds (J.S., L.M.-M.) to share their insights on available evidence. This initiative aims to establish the first steps for critical thinking approaches to address this gap in the scientific literature. 

Although a perspective paper typically does not adhere to the conventional Introduction, Methods, Results, and Discussion (IMRaD) framework, we have structured our manuscript in this format to enhance clarity and facilitate comprehension. A literature search was performed from inception to 2024 in the commonly used scientific databases (MEDLINE, EMBASE, and Google Scholar), adapting the search terms (i.e., «decolonialism», «colonialism», «Indigenous», «cultural competency», «health equity» AND «manual therapy», «osteopathic», «chiropractic») for each database. No limits were placed on language, study design, population, study outcome, or date of publication. Additionally, a snowball sampling technique was used to identify more relevant articles. Validity and quality assessments of the identified papers were not performed. This strategy was used to consider the entire range of information on the decolonial topic in manual therapy. No primary data were gathered (such as through interviews or surveys). The authors of this perspective article adhered to the accepted guidelines for developing commentary [28]. The theoretical foundation for the current manuscript was developed by a working committee comprising professionals with a minimum of 10,000 h of clinical, academic, and research experience (R.Z.-P., F.B., G.D’A., C.L.) [29] as outlined in Appendix A. Findings have been grouped into two subsections. The qualitative data analysis has been discussed in a brainstorming process conducted over 3 online encounters, each lasting 2 to 3 h. 

## 3. Results

The results of the literature search and critical analyses are summarized in two subsections. Section 3.1 provides an overview of the historical roots of Eurocentric scientific dominance in manual therapy. Section 3.2 explores the rationale for the (re)integration of Indigenous body representations, aligned with the body-mind-spirit tenet reintroduced in 2002 by the osteopathic profession, into evidence-informed and person-centered osteopathic care. 

### 3.1. Historical Roots of Eurocentric Scientific Dominance: Descartes’ Body-Machine, European Colonialism’s Legacy, and the Flexner Report

French philosopher René Descartes (1596–1650) argued that the nature of the mind and body are entirely distinct from one another, suggesting that each could exist independently [30]. Cartesian dualism in the 17th century marked a significant ontological and epistemological shift, replacing religious cosmologies with rationalistic frameworks [31]. Descartes’ famous aphorism, “I think, therefore I am”, characterized this transition by prioritizing individual reason over communal experience and relegating the body to a mechanistic construct. Consequently, different dualisms between reason/emotions, man/nature, mind/body, and objectivity/subjectivity emerged because reason took over a religious position with indisputable superiority, thus inferiorizing the body and observing it as a machine, as a set of quasi-perfect pieces and parts [32]. Cartesian dualism not only reified the dominance of reason but also facilitated the objectification of the body as a machine, devoid of subjectivity. Dussel [33] extends this analysis by tracing the origins of Cartesian subjectivity to imperial conquests preceding the 17th century. He argues that the Cartesian “I think” is preceded by the colonial “I conquer”, suggesting that the epistemological centrality of the European subject is predicated on prior acts of conquest and domination. This period marked the era of European colonial expansion, particularly the conquest of America, which started in 1492. European colonization in America was driven by doctrines such as the Inter Caetera papal bulls, which advocated for the subjugation and domination of non-Christian nations [34]. This colonial mindset led to the racial categorization of Indigenous Peoples as “Indian”, positioning them as subjects to be civilized through violent means [34]. Cartesian dualism and the subsequent mechanistic view of the body have significantly influenced epistemological hierarchy in medical education. Eurocentrism perpetuates hierarchies of domination, particularly evident in the realm of epistemology, where it delineates a stark divide between what is deemed valid and valuable, and what is marginalized. Grosfoguel [4] (2013) identifies this phenomenon as epistemicide, wherein holistic cosmologies and Indigenous knowledge systems are erased or delegitimized. De Sousa Santos (2009) [35] further underscores the geographic and gendered biases embedded within Westernized educational canons, exacerbating marginalization and exclusion, highlighting how most of the knowledge in Westernized universities originates from a select few countries, predominantly the United States, England, Germany, France, and Italy. The legacy of European colonialism, embodied in Cartesian dualism and perpetuated by documents such as the Flexner Report, continues to shape medical education and practice. Furthermore, it is important to recognize that this knowledge is frequently influenced by the perspectives of white males, contributing to a lack of diversity in epistemic viewpoints. Indigenous populations continue to experience systemic oppression across cultural, ideological, ontological, epistemological, and subjective dimensions, perpetuating their marginalization outside the dominant epistemic framework [35]. The extent to which such philosophical underpinnings intersect with the legacy of European colonialism, particularly in shaping medical education through documents such as the Flexner Report, remains unknown. However, Grosfoguel (2013) [4] elucidates how Descartes’ formulation engendered a hierarchical division between reason and emotion, and mind, and body, privileging the former at the expense of the latter. 

European colonialism in America has established Western epistemology as the gold standard in medical education, reinforced by Cartesian dualism and the Flexner Report [5]. This report institutionalized the biomedical model, marginalizing Indigenous and non-Western medical traditions, and reflecting broader colonial legacies [36]. The Flexner Report led to a shift towards scientific medical approaches, negatively impacting CAM-oriented institutions. Evidence-based medicine (EBM) further embedded this scientific framework, often excluding patient preferences and practitioner experience [37,38]. Understanding these historical processes is crucial for promoting evidence-informed, person-centered, and culturally sensitive care, as initially advocated by the osteopathic profession. Identifying the roots of scientific monoculture can help address diverse patients’ health needs more effectively [24]. 

We suggest that a decolonial approach in manual therapy, specifically to body representations, is imperative for challenging hegemonic structures, fostering inclusivity, and promoting respect for diverse epistemologies. By acknowledging the historical roots of Eurocentric dominance, we can pave the way for a more equitable and pluralistic healthcare system that embraces the richness of human experiences and knowledge. This is exemplified in the field of manual therapy, particularly within the osteopathic profession, where early osteopathic principles were likely influenced by Native American healing traditions [39] and passive manual approaches, as discussed in the recent literature [14,25]. Notably, the osteopathic body-mind-spirit tenet, directly rooted in the Native American legacy, was reintroduced by the profession in 2002 [21]. Although research on the direct impact of the existential/spiritual dimension on pain experiences remains limited, adopting a biopsychosocial–spiritual approach may resonate with both the Indigenous roots of osteopathy and the latest osteopathic principles [21]. This influence has undergone reassessment considering person-centered care [40] foundations, thus informing evidence-based osteopathic practices (Figure 1). This approach would not only incorporate empathetic listening and explore patients’ meaning and purpose related to their symptoms but also integrate pain-related beliefs into a neuroaesthetic–enactive experience [41], promoting body-mind-spirit cognitive behavioral therapy [42].

### 3.2. Moving beyond the Western Musculoskeletal Scope of Practice in Manual Therapy: Reintegrating Indigenous Body Representations in Person-Centered Care

While multiculturalism focuses on embracing and celebrating cultural diversity to promote inclusivity, decolonialism aims to challenge colonial legacies by addressing systemic injustices and advocating for social change. According to Dussel [43], the initial step towards modern civilization should prioritize epistemic justice, entailing a decolonial shift away from Eurocentrism towards the perspectives of the oppressed. Epistemological decolonization seeks to rectify universalist distortions and recognize the limitations of a monoculture of scientific knowledge rooted in colonial legacies that historically debated the humanity and validity of Indigenous knowledge [44]. Therefore, Indigenous peoples play a crucial role in diversifying epistemologies through intercultural dialogue, which expands the boundaries of scientific knowledge [4]. Decolonizing healthcare necessitates a profound shift in perspectives and practices, acknowledging and embracing diverse forms of knowledge and healing practices to reshape power structures and address historical oppression [45]. This entails integrating Indigenous, holistic, and culturally responsive approaches to care, while centering on the agency, autonomy, and lived experiences of patients [46]. 

Decolonization also involves fostering collaborative and empowering relationships between healthcare providers and patients, promoting holistic well-being, preventive care, and health equity, and confronting structural barriers to access and inclusion within healthcare systems [45,46]. In the US, this process was implemented by the National Center for Complementary and Integrative Health (NCCIH), whose mission is to meticulously assess the efficacy and safety of complementary and integrative health interventions through rigorous scientific inquiry [47]. Governmental institutions produce evidence-based data that can guide healthcare decisions, foster research and education in complementary and integrative health modalities, and facilitate the incorporation of proven complementary and integrative health practices into mainstream healthcare environments. Ultimately, the aim is to cultivate more inclusive, culturally sensitive, and socially just approaches to healthcare that honor the full spectrum of human experiences and promote health and healing beyond the Western perspective. However, crucial debates persist, including questioning whether Indigenous knowledge qualifies as science and probably reflecting ongoing attempts to impose unique modes of knowledge and existence [35]. 

For this reason, proposals to transform institutions from “uni-” versal to “puri-” versal have been put forth to decolonize contemporary scientific knowledge, advocating for a shift from hierarchical structures to an ecology of knowledge [4]. In the context of manual therapy, Western-rooted scientific approaches initially emphasized only the physiological and associated clinical effects [48]. However, contemporary evidence challenges the efficacy of solely biology-focused models, prompting a shift towards person-centered models in education and practice [16,17,20]. Recently, there has been an emphasis on the complexity of patients’ pain experiences, which involve intricate neural processes encompassing sensory, emotional, and cognitive components, both theoretically [18,49,50] and with concrete clinical osteopathic proposals that renovate some distinctive elements of the osteopathic profession (e.g., somatic dysfunction and osteopathic treatment of the head region) [39,51,52,53]. Considering this understanding, contextual factors have been integrated into manual therapies to align more closely with patient values and expectations, as they have been shown to significantly impact clinical outcomes [54]. Contemporary clinical care extends beyond the mere application of manual procedures and recognizes the multidimensional facets of therapeutic encounters, with the potential additional benefits of education, exercise, and cognitive/behavioral interventions [55]. Such an integrative perspective considers not only the practitioner’s expertise and techniques but also patient expectations, verbal suggestions, narratives about body representations and interpretations of symptoms, the strength of the therapeutic alliance, and treatment settings that can evoke biological and psychological responses [54]. These factors influence clinical outcomes via placebo and nocebo effects [54] and, in turn, contribute to improved outcomes across various areas including pain management, disability, and patient satisfaction [56,57,58]. Such a contemporary, humanistic, and holistic perspective within the field of health sciences emphasizes the importance of practitioners simultaneously addressing the biological, psychological, and social dimensions of illness, aiming to enhance their understanding and responsiveness to patients’ suffering. 

This progressive shift towards enhancing humanization in medicine through patients’ empowerment—incorporating their subjective experience, valuing the patient–clinician relationship, and assigning new roles to the patient in clinical decision-making—has also developed over time in the manual therapy field, too. This shift has led osteopathic practitioners to question their professional identity [59]. Unlike physiotherapy, osteopathy and chiropractic were initially created as alternative forms of care to the established medical standards [3]. Consequently, the implementation of EBM in manual therapy faced different barriers and challenges in each of these three professions. For example, a recent cross-sectional study examined the attitudes, skills, and utilization of EBM among UK osteopaths and identified perceived barriers and facilitators affecting EBM integration in clinical care [60]. The results indicated that osteopaths generally expressed support for EBM, self-reported moderate-level skills, and typically engaged in EBM-related activities at a moderately low level [60]. Perceived barriers to the utilization of EBM for UK osteopaths include time constraints and the perceived inapplicability of guidelines in MSK care common to other manual therapists, which overlook the uniqueness and relevance of osteopathic principles in guiding patient care [61]. Indeed, the current concern is that a narrow application of EBM may result in the emergence of a monoculture of knowledge production, which tends to ignore other forms of knowledge from different sociocultural groups. This concern also applies to manual therapy for MSK disorder management, the most common complaint of patients seeking manual care by COP practitioners outside the US, who follow common guidelines based on a Western epistemological framework [55]. 

Some groups are now actively advocating the removal of non-Western epistemological frameworks that have historically been used in the osteopathic profession. The process of classification, hierarchization, marginalization, and rejection of different epistemologies is termed epistemicide [4]. In the context of manual therapy, colonial legacies may still manifest in various ways, including the dominance of Western models of assessment and treatment, marginalization of holistic and culturally responsive approaches, and lack of representation and recognition of diverse practitioners and patients [8,62,63]. Additionally, educational institutions and professional organizations supporting this ideology should become more responsible for ethically addressing these legacies within manual therapy curricula and standards of practice, promoting genuine diversity, equity, and social justice. Cultural diversity, encompassing both Western and non-Western influences on body representations and the expression of pain, has been described in rheumatology [64], a field that incorporates MSK care and could be initially explored to reduce disparities in access to care and inadequate treatment options based on diverse patients’ sociocultural health assumptions. To date, there has been a notable absence of research exploring these fundamental questions and concepts in the peer-reviewed literature. Given this gap in manual therapy, this perspective paper has emerged as the most fitting avenue to counterbalance the predominantly Western-centric viewpoints, especially those advocated by COP.

## 4. Discussion

In the following subsections, we present our integrative perspective designed to provide genuine person-centered care capable of integrating patients’ Western and non-Western/Indigenous sociocultural beliefs regarding body representation in health, illness, and disease within a secular Western clinical scenario. Given that this adaptability has historically been central to osteopathic care in the manual therapy field, we introduce an operational clinical decision-making tool, the Cynefin framework, to manage epistemological flexibility and promote evidence-informed, person-centered, and culturally sensitive care while also fostering interdisciplinary collaboration. 

### 4.1. The Rationale for (Re)Integrating Indigenous Body Representations in Manual Therapy

In contemporary healthcare, there is growing recognition of the potential to integrate Indigenous wisdom with Western knowledge to cultivate holistic healing approaches [10]. However, such integration goes beyond mere acknowledgment; it requires a deeper understanding of Indigenous practices and perspectives. By embracing Indigenous body representations in healthcare, we not only honor and respect Indigenous traditions but also unlock new possibilities for healing and well-being [47]. In the specific context of treating patients in manual therapy, discussing a sensitive topic such as the decolonization of body representations should entail a rigorous critique of epistemological frameworks that have historically prioritized symptom-based or disease-focused treatments rooted in Western biomedical paradigms. This paper highlights how unconscious systemic biases, such as colonial legacies and ongoing power dynamics, can potentially influence therapeutic interventions. Its goal is to ethically emphasize the importance of embracing epistemological flexibility in clinical practice to better address the complexity of contemporary clinical scenarios. More specifically, it explores how appropriately trained practitioners can introduce non-Western frameworks within the Western clinical scenario to align better with patients’ values and expectations, as outlined in the EBM paradigm. To this end, we argue that adopting a decolonial perspective of body representation offers an opportunity for all practitioners to begin reflecting on their own biases and assumptions. This has the potential to build culturally competent and inclusive care, ultimately resulting in improved therapeutic outcomes [65]. Interestingly, many Indigenous practices closely align with the contemporary concepts of placebo effects and nonspecific therapeutic factors [50]. Indigenous healers have long recognized the importance of using body connections and tailoring treatment approaches to address the unique needs of each individual. Western practitioners could take practical steps by incorporating cultural humility, engaging in ongoing professional education and training in diversity and inclusion, and collaborating with community members and organizations. The reintegration of Indigenous body representations in manual therapy represents a critical paradigm shift towards recognizing and valuing the diverse cultural backgrounds and healing traditions of various patient populations, as well as acknowledging various sociocultural belief systems regarding health. 

In osteopathic care, this approach goes beyond mere acknowledgment to actively incorporate cultural nuances into treatment strategies [24]. Indeed, dynamic interactions between the body, mind, and spirit, derived from Native American healing practices, have been reincorporated to guide osteopathic care [14,23,25]. However, knowledge sources that include this existential/spiritual dimension should be carefully contextualized to specific sociocultural environments. This ensures that they can be introduced within secular clinical scenarios, aligning with patients’ values and expectations—the pillars of EBM—to reinforce the therapeutic alliance [14]. 

The most important international professional documents, such as the World Health Organization (WHO) recommendations for osteopathy training [66] and the Osteopathic International Alliance’s report on the status of the global osteopathic profession [12], advocate person-centered interventions that include this existential/spiritual domain through the body-mind-spirit osteopathic tenet. Therefore, osteopathic practitioners may play a crucial role in this process by challenging colonial legacies and advocating for equitable healthcare policies and practices, not only for marginalized and underserved communities [67] but also from a more global perspective. More specifically, returning to integrative frameworks could redefine the scope of practice in osteopathy, emphasizing a holistic approach to health that extends beyond the COP’s focus on MSK care. With its Indigenous legacy combined with current evidence-based practices, the osteopathic profession appears uniquely positioned to develop a robust scientific model of holistic care. Following the emphasis on sociocultural influences on body representations, a logical starting point would be the exploration of the neuroscience of body awareness and its influences on manual therapy practices.

### 4.2. Body Awareness in Manual Therapy: The Neuroscience of Perception, Placebo, and Sociocultural Influences

From a neuroscientific perspective, body awareness refers to the conscious perception and understanding of one’s body and its various parts as well as the ability to sense bodily sensations, movements, and positions. This includes awareness of bodily sensations such as touch, temperature, pain, proprioception (the sense of the body’s position and movement), and interoception (the sense of the body’s internal state, including hunger, thirst, heartbeat, and visceral sensations) [68]. Body awareness emerges from the coordinated activity of multiple brain regions involved in processing sensory information, generating motor commands, integrating internal and external signals, and maintaining a coherent representation of the body in space. Dysfunction or impairment of these neural circuits can lead to alterations in body awareness, as observed in certain neurological or psychiatric conditions [69]. Bravo et al. [70] recently explored the effectiveness of movement and body awareness approaches, such as Tai Chi Chuan, Qigong, and yoga, as adjunct treatments to the usual care for people suffering from fibromyalgia. These therapeutic approaches involve the whole person, encompassing physical, psychological, physiological, and existential/spiritual perspectives, and focus on breathing, postural balance, and awareness. The authors found beneficial outcomes related to pain, pain threshold, number of tender points, sleep quality, fatigue, anxiety, depression, and quality of life. This study expands the Western-centric scope of practice in pain management and musculoskeletal (MSK) function advocated by the COP by incorporating gentle movements within a holistic framework. This approach aims to increase awareness of how the body functions, behaves and interacts with others. With this knowledge inherited from early practitioners and renewed today, osteopathic practitioners have recently proposed additional treatment strategies administered not with patients lying on the table receiving passive manual therapy, but while performing functional movements [71]. The rationale is that hands-on experiential bodywork influences self-tracking of the lived body through body awareness. Non-Western epistemological frameworks can also incorporate current scientific knowledge together with diverse sociocultural beliefs. Van Elk and Aleman [69] introduced a model based on predictive processing (PP) theories, suggesting that existential beliefs arise from how our brains weigh interoceptive and exteroceptive information. 

Multisensory integration combines data from different senses to form a coherent perception of oneself and one’s surroundings. In this framework, our brain’s predictions, informed by past experiences, adjust its processing based on the disparity between expected and observed sensory outcomes. Changes in this process of multisensory integration may result in an altered perception of the body, as observed in out-of-body experiences and psychiatric disorders such as depersonalization and derealization [69]. According to this model, existential/mystical experiences may emerge due to (1) a differential weighting of interoceptive signals compared to exteroceptive signals, (2) changes in the monitoring process of interoceptive or exteroceptive errors, and (3) individual differences in practice, brain structure, function, and development related to interoception. Overall, this framework shifts bodily experiences historically embedded in beliefs about physiological processes, making them a subject for robust scientific appraisal. In following this objective, Maij and van Elk [72] utilized a device described to the participants as capable of producing electromagnetic stimulation of the temporal lobes, which was claimed to trigger spiritual/existential experiences. They observed that highly suggestible individuals and those with a significant history of paranormal experiences were more inclined to report spiritual/existential experiences during sham helmet stimulation. This line of research outlines not only the significant influence of the sociocultural context and prior expectations in shaping existential/spiritual experiences but also the role of placebo and suggestibility, which go beyond the sole neuroscientific explanation. The concept of multisensory integration and existential dimensions, as proposed by Van Elk and Aleman [69], resonates with the body-mind-spirit tenet of osteopathy, inherited from Indigenous healing traditions [14]. This suggests that the ‘spirit’ component could be subject to scientific investigation and play a role in processes such as cognitive reappraisal of pain’s emotional impact and context-dependent analgesia [73]. The interconnectedness of the body, mind, and spirit involves neurophysiological, motor, and cognitive processes that influence emotional responses [70]. Indeed, emotions are mediated through interoceptive and proprioceptive inputs to the brain, potentially influencing manual therapy’s role in regulating emotions via motor behavior and proprioception. Dysfunctional body awareness may therefore contribute to psychosomatic disorders and the perception of persistent physical symptoms [74,75]. Emerging evidence from predictive processing theory supports expanding the scope of manual therapy beyond MSK care, particularly focusing on body awareness, which is crucial for overall health. This suggests that manual treatment could address various components of body awareness within different epistemological frameworks, including Indigenous perspectives [76]. In Native American traditions, a balance among context, mind, body, and spirit, symbolized by the Medicine Wheel, is essential for well-being [14]. 

Touch-based therapies, such as laying on hands, are used within these Indigenous frameworks to promote health and well-being beyond MSK issues, focusing on adjusting the whole person within an existential/spiritual dimension. For example, a form of laying on of hands in Brazil, known as a ‘Spiritist passe’, is described as an exchange of fluids and energies derived from the Spiritist healer, good Spirits, or a combination of both [77]. This passive manual approach, embedded in an existential/spiritual framework, was investigated among Brazilian cardiovascular inpatients (n = 41) and compared to sham and no intervention and appeared to be effective in reducing anxiety levels and the perception of muscle tension, consequently improving physiological parameters [77]. Anthropologists have studied other Indigenous healing practices that extend beyond the mere application of specific techniques. They have described how induced states of hyper-suggestibility during healing ceremonies can serve as powerful tools for belief transmission, enculturation, and social affiliation within a specific group [78]. Chen et al. [79] manipulated practitioners’ expectations in a simulated clinical interaction and observed that these expectations directly influenced patients’ subjective experiences. 

According to the authors, their results provide evidence of a socially transmitted placebo effect, highlighting how healthcare providers’ behavior and cognitive mindsets can impact clinical interactions. Therefore, they suggested training to develop practitioners’ psychological traits such as empathy and communication skills [79]. We argue that osteopathic care could be described within similar frameworks aiming to elicit patients’ top-down placebo responses through therapeutic alliance and setting, embedded in therapeutic rituals, while simultaneously treating the patient via bottom-up manual approaches, i.e., by influencing the somatosensory system. This further highlights the crucial role of patients’ values and expectations, which are shaped by sociocultural influences, as well as the expectations of practitioners. This may be particularly true for whole-body manual osteopathic techniques, such as MSK regional approaches, including cranial and myofascial, as well as visceral techniques. These techniques, usually applied to rested patients in a quiet environment, are similar to laying on hands in other sociocultural environments and are likely to induce positive health outcomes among highly suggestible people. In such clinical scenarios, the (re)integration of Indigenous body representations to promote epistemological flexibility could be useful in updating our models of practice. While the COP perspective sets clear boundaries in its scope of practice, there is an important missing piece to provide ethical care within an evidence-informed framework for patients who have historically requested osteopathic treatments beyond MSK care. While the COP perspective sets clear boundaries within its scope of practice, there is an important missing piece to provide ethical care within an evidence-informed framework for patients who have historically requested osteopathic treatments beyond MSK care. The (re)integration of Indigenous body representations might also be an important step to consider for addressing these patient needs. Finally, such endeavors will align with patient protection efforts, ensuring that they receive safe and ethical care rather than discontinuing existing practices due to outdated supporting models. As we raise awareness of epistemological flexibility by facilitating a contemporary application of the body-mind-spirit osteopathic tenet in specific clinical scenarios, it is important to note that this tenet does not necessarily need to be applied in every clinical situation. Instead, it may serve as an additional important tool when facing complex clinical situations. Within the specific context of this paper, the proposed decolonial perspective entails the (re)integration of Indigenous interpretations associated with known physiological processes involved in body awareness, which are now amenable to scientific inquiry from a PP perspective. These physiological processes, including placebo effects, are largely shaped and influenced by patients’ sociocultural beliefs regarding their health. This is common among all manual therapy professionals who encourage the promotion of patient autonomy and collaboration in healthcare decision-making.

### 4.3. Common and Distinctive Characteristics of Osteopathic Care and Its Relationship to Conventional Healthcare

Some authors in the COP professions advocate breaking down professional boundaries in manual therapy which would be artificially upheld by professional ideologies, and instead promote a form of ‘agnostic’ person-centered and evidence-informed MSK care [8,62,63]. We feel that such a Eurocentric view of knowledge production to guide patient care, which openly dismisses any other origin of knowledge, such as Indigenous, aligns perfectly with an epistemicide agenda reported in other areas of healthcare [38]. This proposal is quite disturbing, since the osteopathic profession—clinicians, educators, and researchers—has always demonstrated flexibility by regularly revising osteopathic principles in 1917, 1953, and 2002 to adapt to evolving healthcare environments, scientific evidence, and patient values and expectations. Assuming the profession would not be able to revise its principles, and would instead suggest the removal of osteopathic practices such as cranial and visceral techniques can be considered, at best, intellectual laziness. Even more problematic is the proposed pan-professional framework’s inability to address patients’ values and expectations shaped by both non-Western and Western sociocultural beliefs, aspects that osteopathic practitioners have addressed over the past 150 years [24]. Nevertheless, the implementation of EBM in the medical environment has further expanded this ongoing process of evolution, challenging historical models of practice with the current evidence [80]. Currently, the osteopathic profession certainly needs to clarify its distinctive clinical features when applying person-centered manual care within a professional interdisciplinary environment. Recent proposals have promoted person-centered osteopathic care, including treatment planning, evidence-informed interprofessional symptom-oriented approaches, and personalized strategies with preventive and health-maintenance aims [81]. Evidence-informed approaches also emphasize collaboration among healthcare professionals from various disciplines to provide whole-person healthcare and address patient care needs. By focusing on the patient’s clinical context and symptoms, interprofessional teams can collectively assess and manage complex healthcare situations more effectively. Shared interprofessional strategies involve leveraging similar skills and objectives across different healthcare disciplines to provide cohesive and coordinated care [82]. However, while the development of an updated model of osteopathic practice in an evidence-based environment should be open to incorporating experiences from other healthcare professionals, it remains crucial to determine the distinctive focus of osteopathic care in addressing health needs and ensuring the quality of health services [83]. To better understand contemporary osteopathic principles and related practices, Handoll [84] analyzed the development of osteopathy in the UK. He wondered whether the osteopathic principles would retain ongoing value in future healthcare or if osteopathy would be categorized as manual therapy alongside physiotherapy for treating defined conditions. According to Handoll [84], while conventional healthcare aims to identify what is wrong, osteopathic care seeks to verify why it is incorrect. It focuses not only on symptoms but also on the whole person, aiming to prevent barriers to health, promote individual wellness and agency, and facilitate recovery from illness. According to current osteopathic principles, and considering health needs, Italian law has recently recognized and regulated osteopathy as a healthcare profession that provides interventions for the prevention and maintenance of health and the prevention of disease [85]. A panel of osteopathic co-authors, clinical experts, academics, and researchers from nine European countries, where osteopathy is recognized and where it is not, discussed the Italian professional profile [85]. Despite the authors declaring positive developments in the legislation for osteopathy, they also reported that the role of osteopaths in health promotion and prevention, as outlined in Italian law, raises critical points regarding the lack of a disseminated model to guide osteopathic care [85]. Therefore, the role of osteopathic care in prevention should be defined within a collaborative framework. According to the WHO *Glossary of Terms* [86], primary prevention aims to prevent disease before it occurs by addressing risk factors in the population. Secondary prevention involves the early detection and treatment of diseases to prevent disease progression or complications. Tertiary prevention focuses on managing existing conditions to minimize disability and improve quality of life. These levels work together to promote health and prevent disease across populations. Personalized tailored treatment with primary, secondary, and tertiary prevention roles encompasses a comprehensive approach to healthcare delivery. Moreover, this personalized treatment approach includes functional bio-psychosocial preventive and health maintenance objectives that are tailored to an individual’s specific needs and circumstances [87]. These objectives were formulated based on the results of the osteopathic evaluation, which assessed the patient’s physical, psychological, and social well-being. Shared decision-making between the patient and osteopathic practitioner guides the development of treatment plans, ensuring that interventions align with the patient’s preferences and goals. Importantly, this personalized approach emphasizes the unique effects of touch inherent to osteopathic practice, that is, interoceptive manual approaches, which contribute to holistic health management. It is important to note that this comprehensive treatment approach requires specialized training and expertise unique to osteopathic practitioners. Osteopathic practitioners possess a nuanced understanding and employ the application of osteopathic principles and techniques, making it challenging for non-osteopathically trained practitioners to deliver effective care. The profession lacks a conceptual and operational framework that provides a comprehensive model to understand all the clinical scenarios encountered in osteopathic practice. Additionally, it cannot propose a person-centered care program that integrates standardized evidence-informed approaches for different clinical contexts along with more personalized approaches focused on individual adaptive capacity. 

Furthermore, no framework has been specifically designed to integrate the most recent osteopathic basic tenets [21], which combine three Western principles with one Indigenous principle, considering a person as the product of dynamic interaction between body, mind, and spirit. To address these challenges, Lunghi and Baroni [88] introduced the Cynefin Framework (CF), not only to reflect the historical diversity of manual techniques that contributed to building a strong professional identity in osteopathy but also as an operational clinical decision-making tool to guide osteopathic care.

### 4.4. The Cynefin Framework, a Clinical Decision-Making Tool That Promotes Epistemological Flexibility in Osteopathic Care

Cynefin, pronounced ‘kuh-nev-in’, is a Welsh word that signifies the multiple, intertwined factors in our environment and our experience, i.e., how we think, interpret, and act, which influence us in ways we can never fully understand [88]. Emerging from research conducted in the fields of complex adaptive systems theory, cognitive sciences, anthropology, narrative models, and psychology, CF was proposed by Tyreman as a tool for managing complexity levels in osteopathic clinical practice [89]. In this perspective paper, CF is also proposed as a clinical tool allowing the reintegration of Indigenous body representations in the contemporary osteopathic management of common complex clinical scenarios. It also incorporates the neuroscientific perspective of body awareness as a physiological basis to support the use of the CF in guiding osteopathic clinicians to deliver person-centered and culturally sensitive care [24]. The CF represents four domains and a confused space, showing the relationship between the individual, experience, and context, which all enable new approaches to decision-making processes in complex environments, such as the relationship between patient and osteopath. The four domains of the CF provide insight into current approaches to osteopathic person-centered care, whether evidence-informed symptom-based, or personalized adaptive approaches (Figure 2).

Practitioners promote symptom-based models in both simple and complicated domains. Conversely, in the complex and chaotic domains, multiple considerations were considered. These include narratives about individual meaning, purpose, and significance related to the self, family, community, and the sacred, expressed through beliefs, values, traditions, and practices, which are integrated during evaluation, diagnosis, and personalized treatment. Moreover, the Cynefin domains illustrate differences in the sequence of practitioner actions and the types of passive manual and active-participative approaches that would make sense for patients according to their specific values and expectations (Table 1).

Overall, the CF can guide practitioners in managing patients who hold Western or non-Western belief systems, thereby reintegrating Indigenous body representations. The CF fosters epistemological flexibility by extending beyond the COP Western perspective in manual therapy focused on MSK pain and function, describing the ability of practitioners to adapt to a wider variety of patients’ beliefs, knowledge, and understanding in different contexts. It enables individuals to navigate complex or uncertain situations more effectively by drawing on diverse ways of understanding the world [93]. Osteopathic practitioners need flexibility to make sense of clinical scenarios where unclear actions and narratives align with the confused domain of the CF. Confused space helps prevent over-identification with the practitioner’s mind and knowledge [23,24,88]. It is time to understand the importance of a patient’s intuition in clarifying the process of integrating body function with the existential/spiritual domain [88]. On one hand, a confused space represents an irresolvable internal contradiction regarding unexplained body representations. On the other hand, it can serve as a safe contemplative space, allowing patients to self-reflect on difficult situations, remain suitably undecided, and meditate. This facilitates connecting with potential solutions that patients feel are viable, thereby initiating their integration into the context and environment [23,24,88]. The confused domain represents the starting point for creating a synchronous relationship and a more effective therapeutic alliance. Both the patient and practitioner could remain suitably undecided and meditate, favoring a connection with what the patient feels is a possible strategy to start. The sense-making process requires unclear elements to be evaluated in light of different domains. Owing to the negotiation process facilitated by the practitioner, the patient was invited to reformulate the confusing elements of his health [23,24,88]. Consequently, the patient could reformulate confused things, shifting them to other domains (i.e., simple, complicated, complex, and chaotic), and applying different reasoning. To better apply the CF in clinical encounters, osteopathic practitioners can use a sheet of paper, blackboard, or screen to graphically represent the sense/decision-making process and facilitate patients’ understanding of their health processes, their disorders, the treatments carried out previously, and the management proposals that make sense, to meet their socio-cultural assumptions and preferences [88].

A simple domain is considered when a single correct answer is perceived to be clear and repeatable in the osteopath–patient dyad to address the patient’s expectations. A linear cause–effect relationship can be observed to sense, categorize, and define a symptom-based approach that the patient can understand well through anatomical–physiological explanations. Practitioner reasoning considers anatomical structures and functions to define manipulative approaches and mobilizations applicable to regional interdependence (RI) [91,92]. RI is a term used to describe clinical observations believed to exist between regions of the body, particularly in the management of MSK disorders proposed in physical therapy [94] as well as in the COP pan-professional approach [91,92].

A complicated domain is assumed when osteopath–patient sense-making relies on expertise or analysis to clarify unclear patient assumptions and address unsolved clinical questions. The cause–effect relationship requires analysis or other forms of investigation or application of specialist knowledge to sense, analyze, and respond by a good-practice approach focused on patient symptoms. In the complicated domain, practitioners follow evidence-informed reasoning to propose patient management strategies based on research results, expert opinions, and clinical experiences in similar biomedical contexts. For example, osteopaths consider approaches suggested by available recent guidelines, like other COP practitioners [95]. A complex domain is recognized when the outcome expected by the patient cannot be predicted and multiple approaches are necessary to identify emergent patterns. 

This involves utilizing both verbal and nonverbal narratives related to body representations to inform the shared sense and decision-making processes between the patient and osteopath. 

In complex scenarios where cause-and-effect relationships are perceived retrospectively, practitioners hypothesize, perceive, and respond with an emergent practice favoring personalized osteopathic approaches. Adaptive minimalist approaches are selected based on the decision-making driver and emergent patterns observed in the patient. This pattern emerges from an agreement between the osteopath and the patient regarding the pleasant perception resulting from touch-based stimuli applied to an area of interest for both parties. This induction of positive surprise in the patient’s prior expectations leads to the redesign of predictive models that are more congruent with the patient’s ecological-social context. Ultimately, this process improves the patient’s agency and function [41]. Emergent osteopathic practices integrate various types of minimalist manipulative and patient-active osteopathic approaches (PAOAs) as potential tools for personalized patient management strategies [87]. 

Finally, the chaotic domain was considered when the elements observed in the patient’s verbal and nonverbal narratives related to body representations were volatile. Consequently, it is critical to immediately obtain an emergent pattern. Chaotic situations are characterized by unknowable and unpredictable conditions, and cause–effect relationships are unclear. Consequently, practitioners act with novel practices to improve patient adaptive ability, sense patient responsiveness to treatment, and discover an emergent response pattern using a personalized approach [24,88]. Osteopathic practitioners should use maximalist approaches, such as interoceptive touch-based mindfulness strategies, close operator–patient proximity, and nonverbal behaviors, to evoke patient patterns and enhance psychological flexibility and self-regulation functioning. Different osteopathic manipulative techniques can be proposed to patients as manual procedures that influence their bodily perceptions within a specific epistemological framework shaped by their primary sociocultural health assumptions [24]. Along with these passive manual therapies, various types of PAOAs have been described as potential tools within maximalist management strategies [87].

### 4.5. The Cynefin Framework to Guide Sense and Decision-Making Processes: A Clinical Vignette

The proposed clinical vignette describes a scenario in which a patient requires osteopathic care despite not experiencing any physical symptoms. The aim is to outline important clinical and ethical considerations that typically necessitate epistemological flexibility, introduce patient-active and passive manual approaches, and establish a therapeutic alliance that would best suit the patient’s values and expectations. This approach aligns with the three pillars of EBM and illustrates how the CF could facilitate shared decision-making in manual therapy scenarios, as requested by osteopathic patients over the past 150 years.

#### 4.5.1. Clinical Scenario

Ricardo was a 44-year-old man who grew up in urban South America and has been living in Europe for the past 12 years, working as a yoga instructor. In addition, he launched a podcast to promote health and well-being. Shortly after moving, he experienced acute episodes of spinal pain. He also experienced irritable bowel syndrome, anxiety, sleep disturbances, widespread pain, and fatigue. The patient had a four-month history of recurrent pain. The general practitioner did not consider it to be attributable to an underlying disease process or structural lesion, and finally defined it as chronic primary low-back pain. Ricardo reported having undergone medical investigations, including colonoscopy, MRI, and seasonal blood tests, all of which yielded no findings. 

He received physical therapy with manipulation and pain-education counseling with psychological support due to the challenges of living far from his family in a different environment. Ricardo also reported being familiar with some specific Native healing practices used within rural communities, such as physical remedies with massages and plant medicine, such as Theobroma Cacao [96], which helped treat physical ailments and better manage psychological distress. In addition, he mentioned breathing and meditative exercises with music as useful tools for stress management. These practices were described as a way to connect with his ‘inner being’ and improve his ‘internal power’ according to Native culture. These are powerful tools for social affiliation within a specific group [78]. He discovered these Native healing practices while traveling in the countryside of his country to interview a medicine woman for his podcast. He also reported low conviviality because he visited his family once a year. Ricardo reported that due to pain, he feels uncomfortable teaching yoga at this stage of his life. Seeking relief after another episode of nonspecific acute exacerbation of spinal pain, Ricardo visited an osteopath who, unlike previous practitioners, suggested a preventive session three months after being symptom-free to assess MSK function and associated psychosocial triggers, and to check if a patient-active approach was incorporated. The following clinical vignette outlines potential practitioner actions—narratives, passive-manual, and patient-active approaches—that would better align with Ricardo’s values and expectations to facilitate shared decision-making in such a complex clinical scenario (Figure 3).

#### 4.5.2. Building a Strong Therapeutic Alliance: Exploring Ricardo’s Expectations and Health Assumptions in a Manual Therapy Scenario to Evaluate the Most Appropriate Narratives of His Body Representation

The implementation of the CF in sense-making processes and clinical reasoning involves the “Two-Eyed Seeing” proposed by Mehl-Madrona [25] to help both patients and practitioners hold a holistic view and to consider the role of psychological and existential domains in health while simultaneously applying biological and biomedical aspects to make shared clinical decisions. The “Two-eyed Seeing” concept originates from Indigenous knowledge systems, especially prevalent among the Indigenous peoples of North America. It embodies the capacity to view the world through both Indigenous and Western perspectives, either concurrently or interchangeably, in different areas such as education, healthcare, and social justice, where diverse perspectives and knowledge systems are necessary for holistic problem-solving and decision-making to address complex issues [25]. This concept illustrates epistemological flexibility in the medical field and could be the most suitable framework for addressing Ricardo’s values and expectations. It incorporates Indigenous body representations and ways of knowing, understanding, and interacting with the world, alongside Western scientific knowledge. Within this framework, the body represents a vehicle for various human experiences and connectedness with others and the environment, therefore requiring optimal function associated with preventive care, that is, not just addressing MSK pain but also potential impacts on MSK function following psychological, and emotional, or existential stressors (Appendix B). Such narratives of body representations would better address Ricardo’s expectations in this prevention-oriented scenario.

### 4.6. Promoting Epistemological Flexibility vs. Applying Scope of Practice: Ethical Considerations

The “Two-Eyed Seeing” framework, blending Western and Indigenous epistemologies, helps clinicians appreciate the wisdom of the Indigenous world alongside contemporary scientific ways of knowing. It was initially introduced to address the needs of specific Indigenous peoples [25]. Such integrative reasoning has been applied to a wider range of populations to address complex clinical scenarios such as medically unexplained symptoms, which have not been fully addressed by Western frameworks. While the incorporation of the existential/spiritual dimension in a secular clinical scenario may foster a more holistic, ethical, and compassionate practice, some practitioners remain reluctant, as this is part of the traditional medicine framework and holds a stereotyped and limited understanding [63]. At the same time, this requirement is part of the professional osteopathic practice standards in the UK, where practitioners are expected to deliver ethical, competent, and safe osteopathic care, considering patients’ needs and values, including religion [102]. Moreover, a recently published qualitative study revealed that Italian osteopaths declared that embracing and applying osteopathic tenets could favor a more inclusive clinical practice [103].

### 4.7. Limitations

This perspective paper has some limitations related to the addressed topic, namely, decolonization and methodological aspects of the manuscript itself. First, we defined ‘decolonization’ in manual therapy as a process of critically examining and challenging historical and cultural biases ingrained within body representations and subsequent care. This choice, involving the acknowledgment and addressing of the ways in which colonial ideologies, power structures, and Eurocentric perspectives have influenced the development and application of professional practices in healthcare, is open to criticism. While such a critical process regarding Indigenous perspectives now appears crucial in contemporary Western care, shaped by evidence-informed practices, it is accompanied by significant controversies that we have chosen to carefully present within the context of manual therapy. Methodologically, the lack of peer-reviewed information on decolonial processes in manual therapy is evident, and it has only recently been suggested in osteopathic care owing to its Indigenous roots [25]. This topic is, at best, ignored by COP authors, despite promoting concepts such as EBM and person-centered care, where patients’ values, beliefs, and expectations should be incorporated into shared decisions. Theoretically, these frameworks should encompass the diversity of body representations and require epistemological flexibility. While we attempted to compassionately present this information, some readers may inadvertently have felt hurt. Our intention is not to blame anyone, but rather to offer different perspectives within the context of a globalized world and avoid perpetuating a cycle of exclusion that has silenced Indigenous epistemologies [104]. 

To address this limitation, we considered diversity in authorship for this paper, involving professionals with Indigenous and Western backgrounds who are familiar with peer-reviewed scientific writing.

## 5. Conclusions

Decolonization involves a concerted effort to address historical injustices and systemic inequalities that perpetuate disparities in healthcare access and outcomes. Indigenous body representations are deeply connected with cultural beliefs, existential/spiritual practices, and traditional healing systems. Those representations were likely embedded in early osteopathic principles resulting from Still’s interactions with Native American tribes in the 19th century. The current body-mind-spirit osteopathic tenet is a legacy of these early principles and could serve as one of the most suitable platforms for integrating Indigenous wisdom into healthcare as a pathway towards more effective, compassionate, and culturally responsive healing practices. 

From a clinical perspective, the Cynefin framework has been introduced to guide decision-making in clinical care, promoting evidence-informed, person-centered, and culturally sensitive care that can go beyond MSK care advocated by the COP pan-professional approach. We propose that the (re)integration of Indigenous body representations into manual therapy can offer three significant advantages: first, it brings social and humanistic value by acknowledging and incorporating diverse perspectives; second, it holds professional value by promoting interoceptive manual approaches; and third, it provides clinical value by potentially enhancing placebo effects within a person-centered framework. Additionally, the sui generis perspectives introduced in this paper are well suited to rigorous research, encompassing biological examinations, such as neuroscientific studies on changes in multisensory integration, and clinical evaluations that capture patients’ experiences through patient-reported outcome measurements. Such research endeavors may contribute to the development of evidence-informed, ethical, and epistemologically flexible frameworks.

## Figures and Tables

**Figure 1 healthcare-12-01149-f001:**
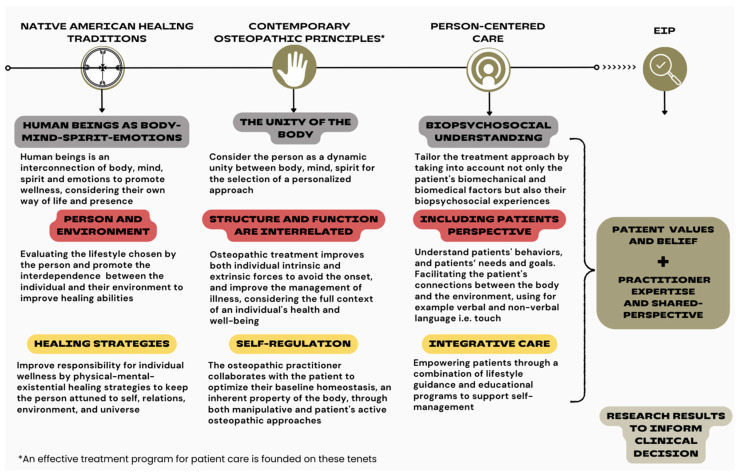
Exploring the intersection of Native American healing traditions and osteopathic principles: insights into person-centered care and evidence-informed practice.

**Figure 2 healthcare-12-01149-f002:**
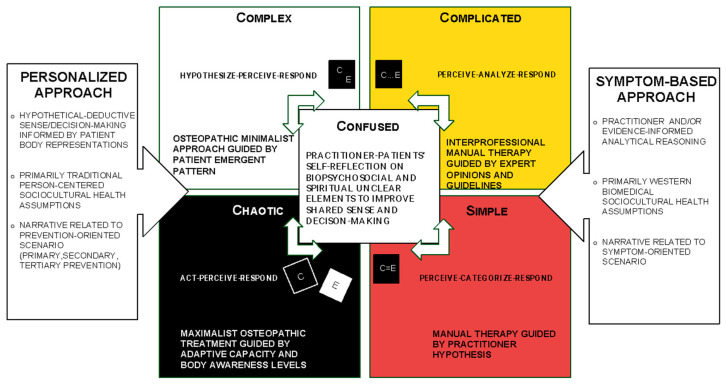
The Cynefin framework and sense decision-making in osteopathic care. Abbreviations: cause (C); effect (E); chiropractic, osteopathy, and physiotherapy pan-professional approach (COP).

**Figure 3 healthcare-12-01149-f003:**
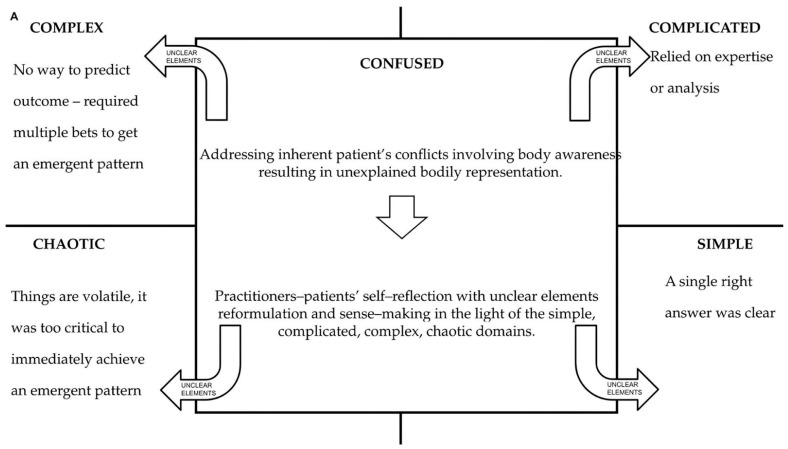
The Cynefin framework guides the sense and decision-making process in a clinical vignette. (**A**). The osteopathic practitioner implements a whiteboard or sheet of paper to graphically represent the sense and decision-making process facilitated by the Cynefin Framework. Practitioners facilitate patients’ self-reflection to simultaneously observe unclear elements of health processes from different points of view and improve sense making. Consequently, the patient could reformulate confused things, shifting them to other domains (i.e., simple, complicated, complex, chaotic), and applying different reasoning. (**B**) The practitioner lists Ricardo’s internal contradictions and unexplained body representations related to chronic primary low-back pain associated with medically unexplained symptoms in the central quadrant of the sheet. (**C**). Thanks to the practitioner’s counseling, Ricardo reformulates confused things, shifting them to other domains (i.e., simple, complicated, complex, chaotic). The patient, being aware of the different reasoning that can be applied, makes sense of precedent treatments and shares with the osteopathic practitioner the decisions regarding the proposed osteopathic treatment planning. The progressive sequence of the overall physiotherapeutic and psychotherapeutic treatment plan is centered on the clinical context, based on available guidelines [95], and is indicated by the circled numbers 1 and 2; the subsequent personalized osteopathic treatment plan defined by shared decision-making [41,51] to promote participative-active osteopathic treatment [71,87] with a combination of touch-based and mindfulness-oriented strategies [97,98,99] and synchronized music listening [100], as well as self-management counseling [101] and is marked by the numbers 3 and 4. Abbreviations: symptom-oriented physical examination (SPE), functional physical examination (FPE), familial symptoms (FS), osteopathic palpatory findings (OPF), structure-function test (SFCT), manual assessment tests of central sensitization (CS), two-point discrimination test (TPD), and Waddell’s sign (WS).

**Table 1 healthcare-12-01149-t001:** Person-centered osteopathic care.

**Personalized Osteopathic Approaches**	
Osteopathic practitioners build a therapeutic alliance to offer possible treatment modalities to patients along with a verbal and nonverbal body narrative that would make sense for them according to their values and expectations. A culturally sensitive narrative, following a neuroaesthetic–enactive experience, introduces how their physical body might influence the complex interaction between body systems [41], i.e., how musculoskeletal function amenable to manual approaches might impact biomechanical, neurological, respiratory–circulatory, metabolic–energetic, and behavioral or biopsychosocial processes involved in individual health [12]. Subsequently, osteopathic practitioners propose maximalist, minimalist, and patient-active participative approaches tailored to individual needs [81].	
Maximalist approaches	Following manual and functional objective examination, osteopathic practitioners can propose maximalist approaches according to patients’ narratives [81]. Maximalist approaches involve different types of passive manual approaches and global body positioning, i.e., preferential fascial patterns to support MSK function and associated biomechanical, postural, neurologic, circulatory, metabolic, and psychological self-regulative functions. For example, one of the maximalist approaches used in circulatory–respiratory and metabolic narratives could be lymphatic pump techniques that implement repetitive passive manual approaches to improve lymphatic flow and the immune system. Another example of the maximalist approach typically used in the neurologic narrative is the total-body fascial unwinding technique, in which touch and meditative–ideomotor movements are supposed to have an interoceptive value. Moreover, whole-body biodynamic osteopathy and other systemic approaches, i.e., osteopathy in the cranial field, are proposed to balance autonomic nervous system networks, resulting in stress reduction.
Minimalist approaches	Osteopathic practitioners suggest a minimalist approach according to body areas of interest considered clinically relevant by both the patient and the practitioner following manual and functional objective examination, with the term ‘somatic dysfunction’ coined to represent a patient-emergent pattern showing a relation between body functioning, patient ability to perform daily activities, and elements of the body framework [81]. Consequently, the osteopath can introduce a neuro–myofascial narrative where active body regions will transmit the biological and physiological effects associated with passive manipulations. The type of touch that is perceived as pleasant by the patient should be considered for the selection of the therapeutic approach [84]. For example, a patient’s pleasant reactivity to rapid compressive touch could guide the selection of high-velocity–low-amplitude techniques, recoil techniques, and patient-active approaches such as rapid stimulation exercises (e.g., using a foam roll and rebound elasticity movement) [81]. Conversely, positive responsiveness to osteopathic slow touch administered along the tangential vectors could influence the choice of indirect myofascial release, with techniques described as balanced ligamentous tension and active-melting stretch approaches [81].
Patient active-participative approaches	Assisted exercise, lifestyle education, empathic communication strategies, and behavioral approaches, based on practitioner–patient proximity and non-verbal behaviors are integrated with minimalist and maximalist manipulative methods to improve biological and psychological adaptability specifically associated with patients’ values regarding their health and well-being [71,87].
**Symptom-Based Manual Therapies**	
Osteopathic practitioners consider the best available evidence and clinical experience within the specific clinical case to administer a symptom-based approach shared by a range of other clinicians such as physiotherapists, occupational therapists, chiropractors, and massage therapists. Many different passive techniques are usually delivered within evidence-oriented manual therapy, and these include soft-tissue techniques, traction, manipulation, and mobilization, often used to treat neurological and orthopedic conditions in combination with other evidence-informed approaches regularly updated in clinical guidelines [90]. Moreover, following a standardized physical examination focused on a patient’s specific location symptoms and anatomical–physiological connected regions, osteopaths administer a standardized manipulative approach for regional interdependence (e.g., upper, bottom, and front quadrant of the body) [91,92].	

## Data Availability

Not applicable.

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
