# Peer review of "Epistemological Flexibility in Person-Centered Care: The Cynefin Framework for (Re)Integrating Indigenous Body Representations in Manual Therapy"

_healthcare, 2024, doi:10.3390/healthcare12111149_

Round 1

Reviewer 1 Report

Comments and Suggestions for Authors

This paper states an important subject that is interesting for readers, but authors should rewrite the method. It isn't clear. Also, each subsection is very long . Unnecessary sentences should be removed.

Comments on the Quality of English Language

It is better that recheck the text. Use familiar words. English very difficult to understand

Reviewer 2 Report

Comments and Suggestions for Authors

Dear authors;

First of all, I would like to thank the authors for their overall efforts during the study. It is a good and clearly described study. The topic is interesting, yet, in its current form, this paper cannot be considered for publication. However, I see value in the research approach and encourage the authors to revise and resubmit their manuscript.

Please find my detailed comments below.

General comments

Overall, this manuscript is well-grounded in existing scholarship and adds to the current landscape. The manuscript was structured well with clear novelty, aims, and methods. The Introduction and discussion are well-written, and the claims are generally supported by the research.

Specific comments

Line 18 (Abstract): Please explain the meaning of (1) just as (2) Methods, (3) Discussion. Introduction / Background?

Line 26: Please try to be more detailed in Methods, considering the Abstract structure and word limit.

I do not understand why the Results were not provided in the Abstract. Please provide the main results of the study.

I am not sure that both the Discussion and Conclusion titles should appear in the Abstract. Please see the journal’s instructions.

Please also re-consider keywords using any number of keywords.

Line 102: Please re-consider reference using. Tuscano et al. (2024) and those similar ones throughout the manuscript.

Line 123: It would be better to start with a new paragraph to present the study purpose. In this way, it would be better for readers.

Line 131: Please be more specific and detailed about the Methods. Most of the readers are not familiar with perspective design.

Line 143: I recommend authors briefly present the study results. There is too much information in the Results section similar to the Discussion.

Line 626: It is very difficult to follow Table 1. Please revise it. I also did not understand the caption of Table 1. This could be text copied and pasted by an AI tool!

Line 836: The Conclusion is too long and detailed. Please present the main findings of the study.

  Comments on the Quality of English Language

 Moderate editing of English language required.

Reviewer 3 Report

Comments and Suggestions for Authors

This is a very interesting article providing both insight and education. It felt though that from line 43 to line 595, was the intro duction and literature review and the study started following that. Also, there are some repetition between the introduction and results' section as they appear. Perhaps changing the chapter headings and integrating thoughts and info of the first part (line 43 to 595) might be able to convey a clearer meaning to the reader. Some comments on the text follow. 

Line 102: the pblication date (2024) is not needed. same comment for line 112. this also appears in the remainder of the manuscript. Please check and correct.

Line 212: perhaps change "was" to "is". I believe it is more suitable

LINE 338: Correct evidecne to evidence

line 478: "According to the authors" as this is a new paragraph, you need to specify to whom you are referring. Consider not starting a new paragraph there.

line 626: check the table's title. 

Comments on the Quality of English Language

Minor English editing required

Reviewer 4 Report

Comments and Suggestions for Authors Dear authors, You address a little-studied topic, with an interesting scope review. Some observations and suggestions:

1) References 32, 43 and 44 are incomplete.
2) In Table 1, it says: "Table 1. This is a table. Tables should be placed in the main text near to the first time they are cited.1" They should put the text. correspondent.
3) Instead of continually using the term "decolonial" (rarely used in PubMed), could you replace it in some sentences with the term "multicultural" (widely used and known by scientific readers, also in PubMed). 4) On lines 489-491, it says: "These techniques, usually applied to rested patients in a quiet environment, are similar to laying on hands in other sociocultural environments, and are likely to induce positive health outcomes among highly suggestible people". I consider it convenient to clarify the following, for the non-specialized reader: ", although this does not explain all the superior therapeutic effect in the experimental groups treated with manual techniques by COP, compared with sham manipulation, both in humans and animals".

5) Ricardo's example is very illustrative; but could you specify more types of Indigenous Body Representations in Manual Therapy? It seems like a somewhat abstract term and it would be interesting to explain it more or make a table of techniques or examples.

Reviewer 5 Report

Comments and Suggestions for Authors

- Please include a comprehensive explanation of the research process in the methodology section.

- The results show that a literature search and critical analysis were performed. Please provide a thorough explanation of the methodology used in conducting the search and analysis.

- Was the research conducted with approvals from an IRB or similar ethical review board if multiple experts, including R.Z-P., F.B., G.D'A., and C.L., participated?

- Please include a table outlining the characteristics of the experts R.Z-P., F.B., G.D'A., and C.L.

- Do EBM and EBP refer to separate ideas? If they are used interchangeably, please provide clarification.

- More instructions or procedures are necessary to show how the suggested method can be put into practice in an actual clinical environment.

Round 2

Reviewer 3 Report

Comments and Suggestions for Authors

Thank you for taking into account my suggestions. The article now appears to have more sense and flow. I am happy with the changes